# Functional Block of Interleukin-6 Reduces a Bone Pain Marker But Not Bone Loss in Hindlimb-Unloaded Mice

**DOI:** 10.3390/ijms21103521

**Published:** 2020-05-15

**Authors:** Hiroki Wakabayashi, Gaku Miyamura, Nobuto Nagao, Sho Kato, Yohei Naito, Akihiro Sudo

**Affiliations:** Department of Orthopaedic Surgery, Mie University Graduate School of Medicine, 2-174 Edobashi, Tsu, Mie 514-8507, Japan; glak1225@clin.medic.mie-u.ac.jp (G.M.); n-nagao@clin.medic.mie-u.ac.jp (N.N.); katosho@clin.medic.mie-u.ac.jp (S.K.); yo-yo@clin.medic.mie-u.ac.jp (Y.N.); a-sudou@clin.medic.mie-u.ac.jp (A.S.)

**Keywords:** osteoporosis, interleukin-6, hindlimb-unloading, osteoporotic pain, tail suspension, CGRP

## Abstract

Interleukin-6 (IL-6) is widely accepted to stimulate osteoclasts. Our aim in this study was to examine whether the inhibitory effect of IL-6 on bone loss and skeletal pain associated with osteoporosis in hindlimb-unloaded (HU) mice in comparison with bisphosphonate. Eight-week-old male ddY mice were tail suspended for 2 weeks. Starting immediately after reload, vehicle (HU group), alendronate (HU-ALN group), or anti-IL-6 receptor antibody (HU-IL-6i group) was injected subcutaneously. After a 2-week treatment, pain-related behavior was examined using von Frey filaments. The bilateral distal femoral and proximal tibial metaphyses were analyzed three-dimensionally with micro-computed tomography. Calcitonin gene-related peptide (CGRP) expressions in dorsal root ganglion (DRG) neurons innervating the hindlimbs were examined using immunohistochemistry. HU mice with tail suspension developed bone loss. The HU mice showed mechanical hyperalgesia in the hindlimbs and increased CGRP immunoreactive neurons in the L3-5 DRG. Treatment with IL-6i and ALN prevented HU-induced mechanical hyperalgesia and upregulation of CGRP expressions in DRG neurons. Furthermore, ALN but not IL-6i prevented HU-induced bone loss. In summary, treatment with IL-6i prevented mechanical hyperalgesia in hindlimbs and suppressed CGRP expressions in DRG neurons of osteoporotic models. The novelty of this research suggests that IL-6 is one of the causes of immobility-induced osteoporotic pain regardless improvement of bone loss.

## 1. Introduction

The loss of muscle tension and gravity in the bones, especially in the trunk and extremity bones, during immobilization leads to loss of calcium from the bones, leading to osteoporosis [1]. Osteoporosis progresses over time and may show little or no apparent sign of progression until pathological fracture of the bone occurs [2].

Hindlimb suspension of rodents by the tail is a well-established approach for creating ground-based models for microgravity and musculoskeletal disuse that mimic many of the physiological changes associated with space flight as well as bed rest [3,4].

In our recent report, hindlimb unloading by tail suspension demonstrated marked bone loss, osteoclast numerical increase, and mechanical hyperalgesia in the hindlimbs, which are prevented by bisphosphonate (BP) treatment. BP treatment prevents upregulation of the neuropeptide pain marker CGRP (calcitonin gene-related peptide) and acid-sensing receptor TRPV1 (transient receptor potential channel vanilloid subfamily member 1) in the dorsal root ganglion (DRG) innervating the hindlimbs [5]. These results suggest that the acidic microenvironment caused by osteoclasts during bone resorption increases disuse osteoporotic pain in hindlimb unloading model.

However, a previous study reported that hindlimb unloading model by tail suspension reduces osteoblast differentiation, induces interleukin-6 (IL-6) secretion, and increases bone resorption [6,7]. In a prospective clinical study, IL-6 receptor inhibitor (IL-6i) increases the bone marrow density of patients with rheumatoid arthritis who had osteopenia [8]. IL-6 is an inflammatory cytokine with wide-ranging biological effects including the maintenance of bone homeostasis.

In the current study, we aimed to investigate pain-related behavior and bone structure to elucidate the mechanism of osteoporotic pain in hindlimb-unloaded (HU) mice treated with IL-6i in comparison with BP.

## 2. Results

### 2.1. Measurement of Pain-Related Behavior with Von Frey Filaments

The mice were tail suspended for 2 weeks, and all mice were reloaded after a 2-week tail suspension. To evaluate the mechanical hyperalgesia around knee osteoporosis, von Frey filaments was used.

At the start of reload (after tail suspension for 2 weeks), pain thresholds of the hindlimb were significantly lower in the hindlimb bone of HU mice than that of the HL mice (Appendix A).

At 2 weeks after reloading, the paw withdrawal threshold and the 50% paw withdrawal threshold remained significantly lower in the HU group than in the HL group (Figure 1A,B). The HU-IL-6i (HU mice treated with anti-IL-6 receptor antibody) and the HU-ALN (HU mice treated with alendronate) groups significantly improved the paw withdrawal threshold and the 50% paw withdrawal threshold compared with HU group (Figure 1A,B). The paw withdrawal frequency stimulations of 0.4-g, 0.6-g, 1.0-g, and 1.4-g filaments were significantly higher in the HU group than in the HL group, whereas they were lower in the HU-IL-6i and HU-ALN groups than in the HU group (Figure 1C).

These results suggest that pain-related behaviors were significantly worse in the HU group than in the HL group, and they were significantly improved in the HU-IL-6i and HU-ALN groups than in the HU group. IL-6 receptor inhibitor and ALN improved mechanical hyperalgesia in hindlimbs induced by unloading.

### 2.2. Immunohistochemical Analysis in the DRGs

Since immobility-induced bone pain was decreased by treatment with IL-6 receptor inhibitor and ALN, we determined if sensory nerves excitation is also reduced in the treated mice by assessing the expression of CGRP in DRG. CGRP is a widely used as neuropeptide marker of pain [5]. 

In the immunohistochemical analysis, the percentage of CGRP-immunoreactive L3, L4, and L5 DRG neurons was significantly increased in the HU group compared with the HL group. It was significantly decreased in the HU-IL-6i and HU-ALN groups compared with the HU group (Figure 2A–D).

### 2.3. Analysis of Three-Dimensional Bone Structure by Micro-Computed Tomography (μCT) 

To determine whether immobility induced osteoporosis around the knee, we evaluated and analyzed bone structure around knees by μCT. 

At the start of reload (after tail suspension for 2 weeks), the HU group had osteoporotic change and significantly decreased bone volume (BV)/tissue volume (TV) of the distal femoral and proximal tibial metaphysis compared with the HL group (Appendix A).

At 2 weeks after reloading, the three-dimensional images of the distal femoral metaphysis (Figure 3A) and proximal tibial metaphysis (Figure 3B) showed less cancellous bone in the HU group than in the HL group. Decreased cancellous bone was improved in the HU-ALN group than in the HU group but not in the HU-IL-6i group.

In parallel with the three-dimensional images, μCT analysis of the distal femoral metaphysis and proximal tibial metaphysis showed that BV/TV and Tb.N remained significant osteoporotic change in the HU group compared with the HL group. Treatment with ALN (the HU-ALN group) improved on BV/TV and Tb.N compared with no treatment (the HU group). However, treatment with IL-6 receptor inhibitor (the HU-IL-6i group) did not improve significantly on BV/TV and Tb. N (Figure 3C–F). Tb.Th and Tb.Sp were not almost significant changes in all groups (Figure 3G–J). The differences between the HL group and the HU-ALN group were not significant in all analyses. The HU-IL-6i group showed no effect on bone morphometry compared with the HU group.

### 2.4. Histological Analysis of Hindlimb Bone

Hematoxylin and eosin staining and the tartrate-resistant acid phosphatase (TRAP) method were used to assess histological analysis of bone structure and to identify osteoclasts in hindlimb bone.

The HU group had less cancellous bones in the distal femoral metaphyses and proximal tibial metaphyses than the HL group. The HU-ALN group improved cancellous bone loss compared with the HU and HU-IL-6i groups (Figure 4A). Obvious fractures of the femur and tibia in mice were not found on histological analysis. In parallel with cancellous bone loss, the HU group had the more significant number of TRAP-positive osteoclasts (OC) in the distal femoral metaphysis and proximal tibial metaphysis compared with the HL group. The HU-ALN group treated with ALN had the less significant number of OC compared with the HU group (untreated). The HU-IL-6i group treated with IL-6 receptor inhibitor also had significantly decreased number of OC (Figure 4A–C).

## 3. Discussion

The tail-suspended rat model was first developed at the National Aeronautics and Space Administration Ames Research Center in the mid-1970s to simulate and investigate specific aspects of space flight. The tail-suspended mouse model was first reported in 1992 by Simske and coworkers [4]. The HU model was developed to study skeletal metabolism in space [3]. A major health concern associated with mechanical unloading conditions such as space travel is the promotion of bone loss [9]. Space flight results in reduced bone mass, especially in weight-bearing bones. This is a condition suggested to resemble disused osteoporosis. For example, during spaceflight, astronauts lose about 1% of their bone mass a month. This is 10 times faster than the loss of bone mass after menopause [10]. Long-term disused osteoporosis is not easily reversible. In a primate study immobilized for 7 months, normal bone formation is not observed until 6 months after resumption of activity [11].

Several studies have used a HU rodent model combined with BP treatment to alter bone metabolism in mice [12,13]. Starting treatment with BP immediately after tail suspension, our previous results indicated that the suppressive effect of BP on osteoclast function may contribute to an improvement in skeletal pain in HU model, and BP prevents expressions of CGRP and TRPV1 in DRG neurons [5]. We further investigated whether BP or IL-6i administration after the induction of osteoporosis prevents bone loss and pain-related behavior in osteoporosis model mice with hindlimb unloading.

IL-6 is multifunctional and is involved in the regulation of various physiological processes, including immune response, acute-phase reaction, and hematopoiesis [14]. IL-6 contributes to increased pain and hyperalgesia in inflamed tissue. In this study, IL-6 is suggested to be one of the causes of immobility-induced osteoporotic pain. In a previous report, tail suspension reduces the osteogenic potential of stromal bone marrow cells and already differentiated osteoblasts. Rat hindlimb unloading by tail suspension reduces osteoblast differentiation, induces IL-6 secretion, and increases bone resorption in ex vivo culture [6,7]. IL-6 levels of bone marrow-derived cells are increased in tail-suspended rat compared with control rat. In this study, MR16-1, an anti-IL-6R antibody, suppressed HU-induced mechanical hyperalgesia in the hindlimb and prevented CGRP expression in DRG neurons.

CGRP plays an important role in inflammatory pain and immune responses. In in vitro and in vivo studies, CGRP can increase mRNA and protein levels and can release IL-6 from macrophages in response to various stimuli. [15,16,17,18]. In long-term bone marrow cultures, CGRP stimulates dose- and time-dependent increases of IL-6 in bone marrow-derived macrophages (BMDMs) [18]. These results suggest that BMDM is responsible for CGRP-induced IL-6 in bone marrow. In a previous study, injured intervertebral disc (IVD) showed increased production of IL-6 and IL-6R and increased expression of CGRP in DRG sensory neurons that innervate the injured IVD. Treatment of anti-IL-6R monoclonal antibody suppressed CGRP expression triggered by IVD injury in DRG neurons [19]. Our results and previous reports suggest that facilitating the production and release of IL-6 from invading macrophages and/or bone marrow is one of the major mechanisms associated with the role of CGRP in maintaining bone pain.

IL-6 is widely accepted to stimulate osteoclasts according to previous reports that showed that IL-6 induces the differentiation of osteoclast precursor cells into mature and active osteoclasts, both in vitro and in vivo [20]. The role of IL-6 in bone remodeling in vivo has been studied using various transgenic and knockout animal models. Previous studies reported that young (6–8 weeks old) IL-6-/-mice had no significant bone abnormalities [21,22] or only had higher bone turnover than wild type controls [23]. Although anti-IL-6R antibody did not prevent the loss of bone mass, it partially prevented the increase in osteoclast number in this study. Lazzaro et al. [24] reported that IL-6 trans-signaling inhibition prevents the increase in osteoclast number and trabecular bone loss associated with ovariectomy. However, in the study, both cis- and trans-signaling by IL-6 did not inhibit trabecular bone loss associated with ovariectomy. MR16-1, which we used in this study, inhibits both cis- and trans-signaling (pan-inhibitors) [25]. MR16-1 did not improve trabecular bone loss in the ovariectomized mice in our recent study [26].

The present study had several limitations. As a first limitation, the provided measurements are not a direct measure of bone pain. The previous study reported that HU significantly reduced bone mineral density and muscle mass in the cancellous region of the distal femur based on the ratio of soleus to body mass. However, there was no difference in the soleus, muscle mass, or the muscle mass to body mass ratio of extensor digitorum longus (EDL) between the control and HU groups. Therefore, we think that pain-related behavior in our model participated more in bone than other tissues such as muscle [27]. A second limitation of this study includes the dose- and time-dependent effects of anti-IL-6R antibody. We did not demonstrate how their treatment of anti-IL-6 receptor antibody could sufficiently block the IL-6. Therefore, this may be attributed to the little pharmaceutical quantity of antibody and/or treatment duration for improvement of bone loss. However, in our recent study, anti-IL-6R antibody even at 4-week treatment had no effect on ovariectomy-induced bone loss [26]. We did not investigate the treatment of vehicle group (HL mice). Future studies should address the dose-dependent effects of treatment in hindlimb unloading-induced bone loss and effects of treatment in hindlimb bone of vehicle group. Additionally, more mechanistic studies examining IL-6 and CGRP expression levels in the bone need to be addressed.

## 4. Materials and Methods

### 4.1. Reagents

MR16-1 (rat anti-mouse IL-6 receptor monoclonal antibody) was provided by Chugai Pharmaceutical Co. (Tokyo, Japan). Alendronate (ALN; Bonalon^®^ Bag for intravenous infusion; Teijin Pharma Ltd., Tokyo, Japan) was purchased from Teijin Pharma Ltd. (Tokyo, Japan).

### 4.2. Animals

The present experiments were approved by the Mie University Animal Care Committee (approval number: 27–21, approval date 4 January 2016) and were undertaken in accordance with the ethical guidelines of the National Institutes of Health. A randomized, prospective, controlled animal model design was used. All efforts were made to minimize animal suffering and the number of animals used. Sample size was determined using a power analysis for an alpha of 0.05 and a power of 0.80 using G*POWER3 [28].

Seven-week-old male ddY mice were purchased from Japan SLC (Hamamatsu, Japan) and acclimated for 1 week before the start of the experiment. Experiments were conducted on ddY mice weighing 34–39 g, as in previous reports [5,26]. The mice were housed in individually ventilated cage at a temperature-controlled room (23 ± 1 °C) with a 12-h light/dark cycle (lights on from 7:00 to 19:00) and given free access to food and water. No mice had any abnormal health condition before the experiments.

### 4.3. Hindlimb Unloading

The protocol of tail suspension was modified from that of Nakagawa [5]. The forelimbs were normally loaded, and the movement of the hindlimbs was free without weight-bearing (unloaded). The overall suspension period was 2 weeks. After a 2-week tail suspension, all mice were reloaded and were treated for 2 weeks. In the hindlimb-loaded (HL) mice as the age-matched control group, the active leg was loaded via the swivel. HL mice were housed individually under the same conditions, but they were not subjected to hindlimb unloading (HU).

### 4.4. Experimental Protocol

The mice were tail suspended for 2 weeks and assigned to 2 groups: HL mice with only tail suspension (*n* = 8/group) and HU mice with tail suspension (*n* = 24/group). After a 2-week tail suspension, all mice were reloaded and assigned to 4 groups: HL mice with only tail suspension treated with vehicle (HL group) as the control group; HU mice with tail suspension treated with vehicle (HU group); HU mice treated with anti-IL-6 receptor antibody, a potent IL-6 receptor inhibitor (HU-IL-6i group); and HU mice treated with alendronate (HU-ALN group) (*n* = 8/group). The mice were randomly allocated to treatment and control groups. Immediately after reload, vehicle (physiological saline), ALN, or IL-6i was injected subcutaneously. ALN was injected with 40 μg/kg ALN subcutaneously twice a week for 2 weeks, as described previously [5,10]. According to a previous report, IL-6i was initially injected with 2 mg/mice subcutaneously, followed by 0.5 mg/mice once a week for 2 weeks after 1 week [29].

At the end of the 2-week treatment period, the mechanical sensitivity of the hindlimbs was examined using von Frey filaments. Following the examination, mice were sacrificed with an intraperitoneal injection of pentobarbital sodium (0.5 mg/kg). Bilateral hindlimbs were removed to conduct micro-computed tomography (μCT), and the bilateral dorsal root ganglion (DRG) from L3 to L5 were collected for immunohistochemical analysis.

### 4.5. Measurement of Pain-Related Behavior with Von Frey Filaments

The mechanical hyperalgesia around knee osteoporosis was assessed with von Frey filaments by one observer who was blinded to the experimental group. The mechanical nociceptive threshold of the hind paw was determined as described elsewhere [5,26,30]. The von Frey tests were conducted at the start of reload (after tail suspension for two weeks) and at 2 weeks after reload. The von Frey tests were performed as described previously [5,26]. Before beginning the experiments, each mouse was placed in a separate plastic chamber and habituated to the test apparatus for 60 min before testing. Von Frey filaments (Aesthesio^®^, DanMic Global, San Jose, CA, USA) were applied to the middle of the plantar surface of the hindpaw for about 2–3 s or until the animal displayed a nociceptive response, consisting of paw lifting and/or shaking. To evaluate the frequency of the withdrawal response, five von Frey filaments with forces of 0.4 g, 0.6 g, 1.0 g, 1.4 g, and 2.0 g were applied five times each in ascending order of force. The results were expressed as the percent response frequency of paw withdrawals. To evaluate the withdrawal threshold, each von Frey filament was applied once, starting with 0.008 g and increasing until a withdrawal response was reached, which was considered a positive response. The lowest force producing a response was considered the withdrawal threshold. To evaluate the 50% withdrawal threshold, a series of nine von Frey filaments calibrated to produce incremental forces of 0.02 g, 0.04 g, 0.07 g, 0.16 g, 0.4 g, 0.6 g, 1.0 g, 1.4 g, and 2.0 g was applied through the wire mesh floor of the chamber. Testing was initiated with the 0.6-g filament. In the absence of a clear paw withdrawal response, increasingly stronger filaments were presented consecutively until such a response was elicited. If the 0.6-g filament elicited a response, filaments with decreasing strength were presented until identification of the first one that failed to cause paw withdrawal. Data were collected using the up-down method [31] to calculate the 50% mechanical paw withdrawal threshold.

### 4.6. Analysis of Three-Dimensional Bone Structure by μCT

To determine 3-dimensional bone structure, femurs and tibias were imaged using a μCT scanner (R_mCT; Rigaku Corporation, Tokyo, Japan) at a tube voltage of 90 kV, tube current of 0.15 mA, slice thickness of 20 μm, and pixel size of 20 μm. At the start of reload (after tail suspension for two weeks), mice were imaged under anesthesia, and at 2 weeks after reload (treatment), isolated femurs and tibias of sacrificed mice were imaged using a μCT scanner. The scanned region contained both cortical and trabecular bone in the distal femoral metaphysis and the proximal tibial metaphysis was located approximately 200 μm from the growth plate. Three-dimensional images were reconstructed from these tomograms of 50 slices (1000 μm) in trabecular bone and analyzed using three-dimensional image analysis software (TRI/3D-BONE, RATOC System Engineering, Tokyo, Japan). Osteoporotic evaluation was performed on the basis of bone volume fraction (BV (bone volume)/TV (tissue volume), %), trabecular number (Tb.N, /mm), and trabecular separation (Tb.Sp, μm) [5,26].

### 4.7. Immunohistochemical Analysis of Hindlimb Bone and DRG

After the behavioral tests, the mice were sacrificed and their spines and hindlimb bones were removed. Tissue preparation and immunostaining of sections were performed as described previously [5,26]. Isolated femora and tibias were fixed in 4% paraformaldehyde in 0.1 M phosphate-buffered at 4 °C for 1 day. Decalcification was carried out in 10% EDTA-2Na (ethylene diamine tetraacetic acid disodium) solution (pH 7.4) for 4 weeks at 4 °C. The decalcified tissue blocks were embedded in paraffin and sectioned into 5-μm thick sections parallel to the mice frontal plane. Sections were stained with Hematoxylin and Eosin (H&E) for histological analysis of bone structure. To identify osteoclasts in hindlimb bone, we used the tartrate-resistant acid phosphatase (TRAP) method. Sections were incubated for 30 min at 37 °C in 26 mL sodiumacetate buffer (0.5% *w*/*w* sodium acetate, 0.8% *v*/*v* acetic acid, pH 5.0) containing 15 mg napthol AS-B1, 1 mg Fast Red Salt, 75 mg sodium tartarate, and 0.8 mg sodium nitrite. Following incubation, sections were counterstained with hematoxylin, mounted, and coverslipped. In the distal femur and the proximal tibia, the number of osteoclasts/bone perimeter (N.Oc./B.Pm.) was determined as the number of TRAP-positive osteoclast within an area 0.5 mm in length and 2 mm in width, beginning from the most distal part of the growth plate.

The peptidergic nociceptive fibers were recognized by means of immunoreactive CGRP innervated to bone [32]. Previous studies showed the L3-5 DRG neurons (the primary afferent neurons) innervating the hindlimb bones [33,34]. The samples were fixed in 4% paraformaldehyde in 0.1 M phosphate-buffered at 4 °C for 1 day and embedded in paraffin, and serial 5-μm cross sections were processed. Immunohistochemical analysis of CGRP expression was completed for the L3, L4, and L5 DRG neurons. The primary antibodies used included anti-CGRP antibody (rabbit polyclonal; Sigma-Aldrich, St. Louis, MO, USA). For isotype control, the sections were incubated with rabbit isotype-matched immunoglobulin G (IgG) instead of the primary antibodies. No immunoreactions were observed in the control group. The average percentages of immunoreactive L3, L4, and L5 DRG neuron sections were calculated on immunostained section examination. 

The immunostained sections were reviewed by each one observer who was blinded to the experimental group.

### 4.8. Statistical Analysis

All numerical values are expressed as means ± standard deviation (SD). Comparison between groups were made using one-way ANOVA test and post hoc least significant difference (LSD) test if data were normally distributed. If data were not distributed normally then data were analyzed with Kruskal–Wallis test and post hoc Mann–Whitney test. *p* < 0.05 was considered significant. All statistical analyses were performed using IBM SPSS Statistics (IBM Japan, Tokyo).

## 5. Conclusions

In summary, anti-IL-6R monoclonal antibody did not prevent hindlimb unloading-induced bone loss, but it suppressed mechanical hyperalgesia in the hindlimbs and upregulation of pain marker in the DRG. The novelty of this research is that IL-6 is one of the causes of immobility-induced osteoporotic pain regardless of improvement of bone loss and the results may be useful as basic research on low back pain of long-standing disuse osteoporosis.

## Figures and Tables

**Figure 1 ijms-21-03521-f001:**
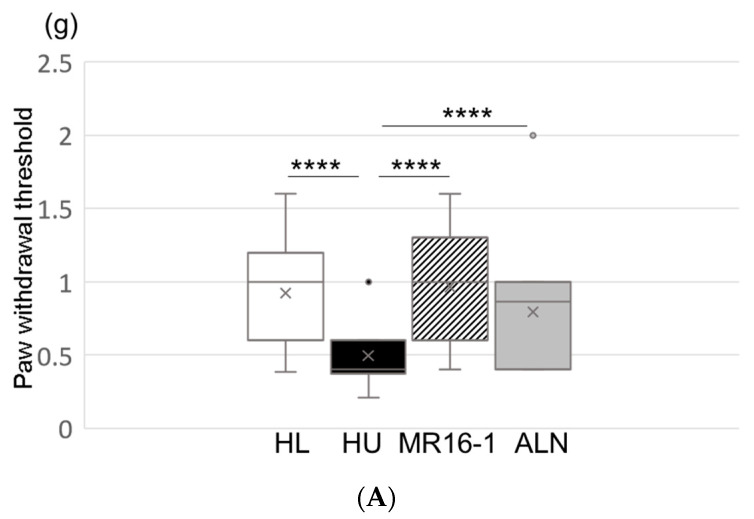
Effect of pain-related behavior by anti-IL-6 antibody and alendronate (ALN) administration in hindlimb-unloaded (HU) mice: (**A**) Paw withdrawal threshold (g). (**B**) 50% paw withdrawal threshold by the up-down method (g). (**C**) Withdrawal frequency stimulation (%). Top, bottom, and middle lines of the graph correspond to the 75th percentile, 25th percentile, and median, respectively. Cross represents mean. Each circle represents an outlier (*n* = 8 in each group). * *p* < 0.05, *** *p* < 0.005, and **** *p* < 0.001.

**Figure 2 ijms-21-03521-f002:**
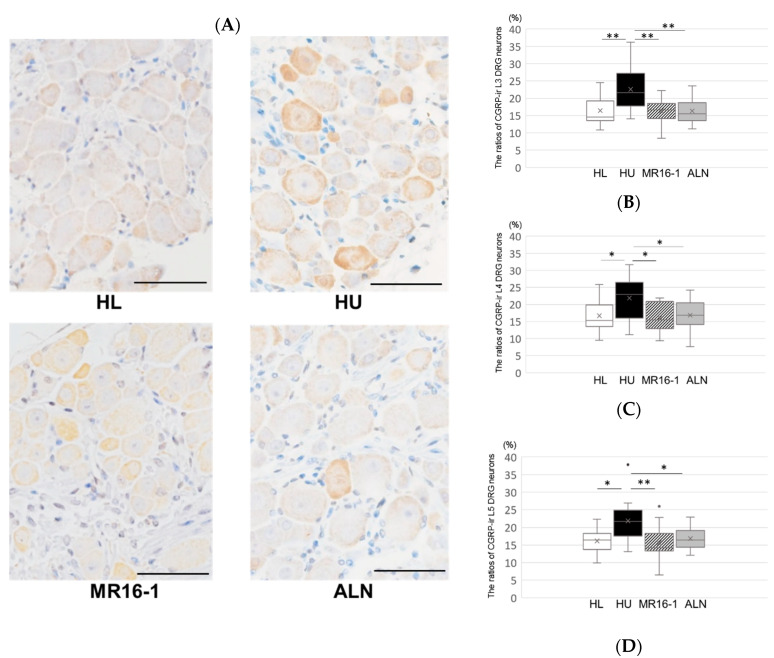
Immunohistochemical analysis of Calcitonin gene-related peptide (CGRP) expression in dorsal root ganglion (DRG) neurons: (**A**) CGRP expression in the DRG neurons (Scale bar is 50 μm). The ratios of CGRP-immunoreactive L3 (**B**), L4 (**C**), and L5 (**D**) DRG neurons (%). Top, bottom, and middle lines of the graph correspond to the 75th percentile, 25th percentile, and median, respectively. Cross represents mean. Each circle represents an outlier (*n* = 8 in each group). * *p* < 0.05 and ** *p* < 0.01.

**Figure 3 ijms-21-03521-f003:**
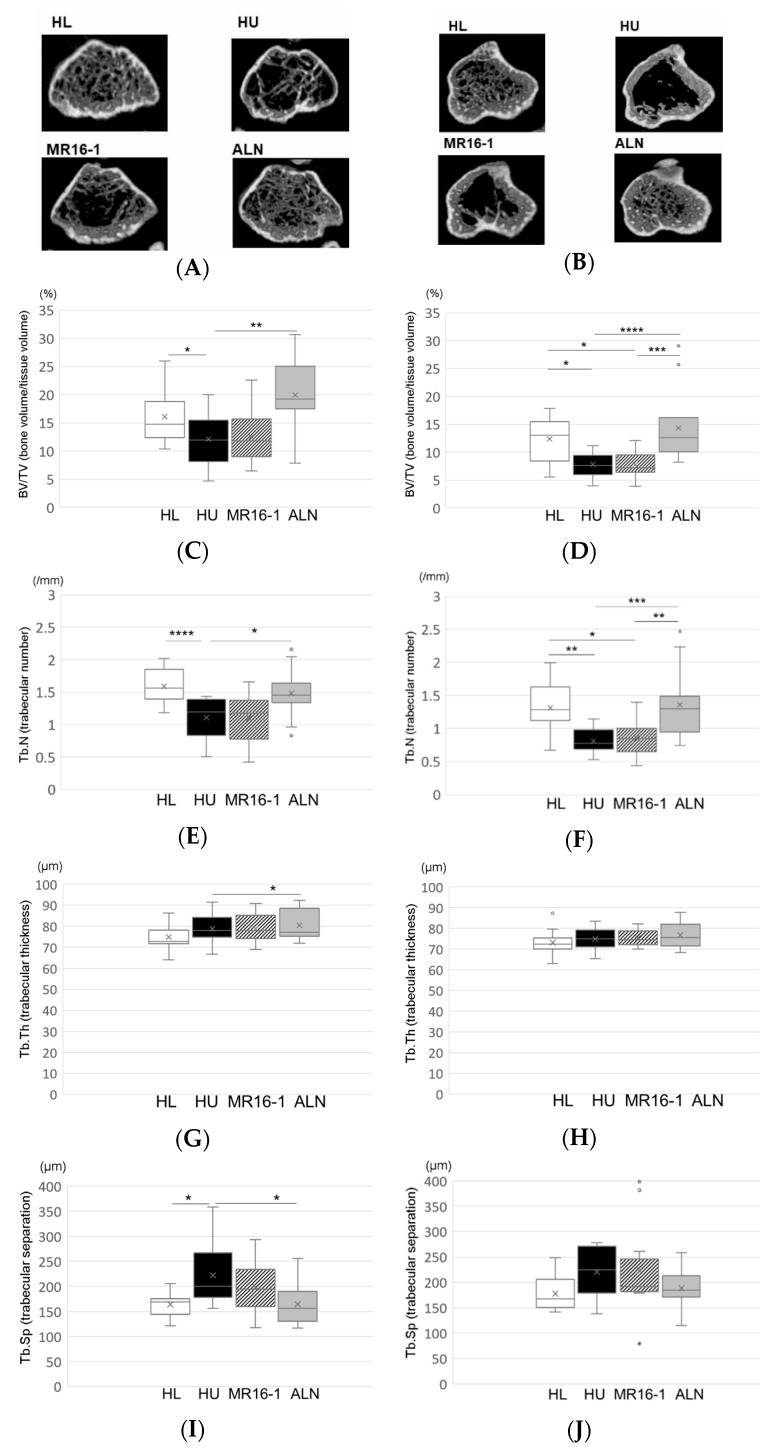
Micro-CT analyses of the distal femoral metaphysis and the proximal tibial metaphysis: Three-dimensional images of the distal femoral metaphysis (**A**) and the proximal tibial metaphysis (**B**–**D**) BV/TV (bone volume/tissue volume) (%), (**E**,**F**) Tb.N (trabecular number) (/mm), (**G**,**H**) Tb.Th (trabecular thickness) (μm), and (**I**,**J**) Tb.Sp (trabecular separation) (μm). Top, bottom, and middle lines of the graph correspond to the 75th percentile, 25th percentile, and median, respectively. Cross represents mean. Each circle represents an outlier (*n* = 8 in each group). * *p* < 0.05, ** *p* < 0.01, *** *p* < 0.005, and **** *p* < 0.001.

**Figure 4 ijms-21-03521-f004:**
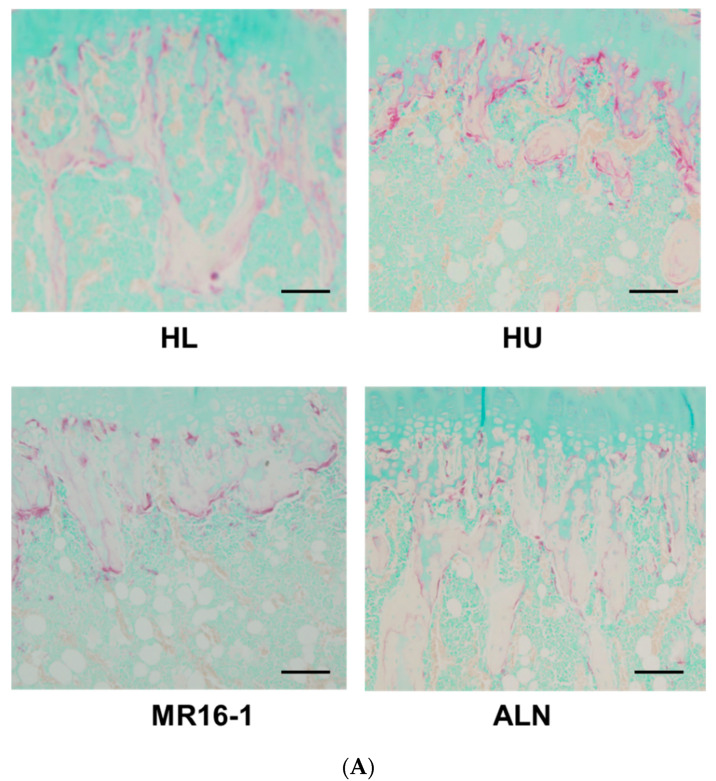
Histological analysis of hindlimb bone: Tartrate-resistant acid phosphatase (TRAP) staining of the proximal tibial metaphysis, Scale bar is 100 μm (**A**). Histological analysis of the number of osteoclasts/bone perimeter (N.Oc./B.Pm) in the distal femoral metaphysis (**B**) and the proximal tibial metaphysis (**C**) (/mm). Top, bottom, and middle lines of the graph correspond to the 75th percentile, 25th percentile, and median, respectively. Cross represents mean. Each circle represents an outlier (*n* = 8 in each group). * *p* < 0.05, ** *p* < 0.01, *** *p* < 0.005, and **** *p* < 0.001.

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
