# Peer review of "Functional Block of Interleukin-6 Reduces a Bone Pain Marker But Not Bone Loss in Hindlimb-Unloaded Mice"

_ijms, 2020, doi:10.3390/ijms21103521_

Round 1
Reviewer 1 Report
The manuscript investigates the effect an anti-IL-6-receptor antibody versus alendronate on bone microstructure, osteoclast abundance and bone pain in hindlimb-unloaded mice. At week 2, the mechanical sensitivity was examined using von Frey filaments and as a pain surrogate marker the CGRP immunoreactivity was investigated in the dorsal root ganglion (DRG). The bone microstructure was investigated by µCT and the abundance of osteoclasts was investigated in TRAP stained sections. The overall conclusion drawn is that the anti-IL-6R antibody did not prevent hindlimb unloading-induced bone loss, but suppressed the investigated markers of pain.
Major concerns:
- The CGRP immunoreactivity in the DRG is really not convincing in the provided images (figure 2). The staining appears more like a shade, than a convincing crisp staining. This renders the quantifications highly questionable. Moreover, the underlying procedure is poorly described. Which controls were used to validate the staining? Which detection and chromogen were used? How was the analysis performed by the three investigators? Was the analysis performed blinded with regard to the condition?
- The analysis of TRAP-positive osteoclasts in figure 4 is not done correct, and the provided images are fare from convincing. Its histomorphometric incorrect to simple report the number of osteoclasts. Whenever reporting a cell profile number, it should always be normalized to the investigated bone perimeter (Oc.N/B.Pm). The differences may simply reflect the difference in BV/TV. This should be redone.
- In general the style of graphs in non-classical and lacking basic information about the reported parameters that are only reported in the figure legends. This should be changes.
- Basic information about the µCT analysis, like the resolution, is missing. Even though they cite a paper, they materials and methods should be clear on its one.
- It’s unclear whether all the investigations were performed blinded? Meaning that the observer was unaware of the condition during the analysis. Now was this secured?
- The number of animals used in each graph/condition should be reported in the figure legend.
- The huge overlap of the bars SD makes one question the reported statistics. Please provide graphs were one can see the individual points for each animal, not just the mean.
- The results would have been more convincing if the authors had included additional measures of bone pain? As the provided measurement are only surrogates of pain, not a direct measure of bone pain.
Minor concern:
In the materials and methods on line 235-238, the paragraph about the origin of the model should be deleted or moved to the introduction or discussion. Not relevant in this chapter.
Author Response
Replies to Reviewer #1
Authors are grateful to Reviewer #1 for encouraging comments. We have revised the indicated parts of the manuscript according to the comments. Corrections in the newly revised manuscript are red color.
Please note that your review comments are shown in italic below and our replies in non-italic.
On the comment of [The CGRP immunoreactivity in the DRG is really not convincing in the provided images (figure 2). The staining appears more like a shade, than a convincing crisp staining. This renders the quantifications highly questionable. Moreover, the underlying procedure is poorly described. Which controls were used to validate the staining? Which detection and chromogen were used? How was the analysis performed by the three investigators? Was the analysis performed blinded with regard to the condition?]
Reply: Thank you for your valuable pointing and suggestions. We described the underlying procedure in detail. We used the appropriate rabbit IgG as the isotype control. We added the following sentences, “For isotype control, the sections were incubated with rabbit isotype-matched immunoglobulin G (IgG) instead of the primary antibodies”, on Line335-336. We provided the higher magnification of groups in CGRP expression in the DRG neurons in supplementary data (F).
The three investigators performed the blinded analysis. The number of reactive cells (TRAP and CGRP) and pain-related behavior were counted and measured by each one observer who was blinded to the experimental group. We changed the following sentences, “The immunostained sections were reviewed by each one observer who was blinded to the experimental group” on Line340-341.
On the comment of [The analysis of TRAP-positive osteoclasts in figure 4 is not done correct, and the provided images are fare from convincing. Its histomorphometric incorrect to simple report the number of osteoclasts. Whenever reporting a cell profile number, it should always be normalized to the investigated bone perimeter (Oc.N/B.Pm). The differences may simply reflect the difference in BV/TV. This should be redone.]
Reply: Thank you for suitable comment. We noted that “the number of TRAP-positive osteoclast was determined within an area 0.5 mm in length and 2 mm in width” on Line325-327. This means the number of osteoclasts/bone perimeter (N.Oc./B.Pm.). We changed “N.Oc./B.Pm” on Line326-329.
On the comment of [In general the style of graphs in non-classical and lacking basic information about the reported parameters that are only reported in the figure legends. This should be changes.]
Reply: Thank you for your valuable suggestions. We changed the style of graphs in figure 1 and 3.
On the comment of [Basic information about the µCT analysis, like the resolution, is missing. Even though they cite a paper, they materials and methods should be clear on its one.]
Reply: Thank you for your valuable suggestions. We added the basic information about the µCT analysis in details on Line303-313.
On the comment of [It’s unclear whether all the investigations were performed blinded? Meaning that the observer was unaware of the condition during the analysis. Now was this secured?]
Reply: Thank you for suitable comment and suggestion. All the investigations were performed blinded. We changed the following sentences, “The mechanical hyperalgesia around knee osteoporosis was assessed with von Frey filaments by one observer who was blinded to the experimental group” on Line280-281, and “The immunostained sections were reviewed by each one observer who was blinded to the experimental group” on Line340-341.
On the comment of [The number of animals used in each graph/condition should be reported in the figure legend]
Reply: Thank you for suitable comment. We noted the number of animals in each figure legend, “n = 8 in each group”.
On the comment of [The huge overlap of the bars SD makes one question the reported statistics. Please provide graphs were one can see the individual points for each animal, not just the mean.]
Reply: Thank you for suitable comment. However, we did not have software that can be displayed dot plot graph (individual points graph). We changed that figures were expressed as box plot. And we added the following sentences in each figure legend, “Top, bottom, and middle lines of the graph correspond to the 75th percentile, 25th percentile, and median, respectively. Cross represents mean. Each circle represents an outlier.”
On the comment of [The results would have been more convincing if the authors had included additional measures of bone pain? As the provided measurement are only surrogates of pain, not a direct measure of bone pain.]
Reply: Thank you for suitable comment. However, to my knowledge, there is no direct measure of bone pain. In previous study, Fluckey et al (Acta Physiol Scand. 2002;176(4):293-300) reported that HU resulted in significant reductions of BMD in cancellous regions of the distal femur and of muscle mass based on the soleus to body mass ratio. However, compared with the soleus, muscle mass or the muscle mass to body mass ratio of extensor digitorum longus (EDL) was not different between the control and HU groups. So, we think that pain-related behavior in our model participated more in bone than other tissues such as muscle. And we added the limitation on Line218-224, “the provided measurements are not a direct measure of bone pain. The previous study reported that HU resulted in significant reductions of bone mineral density in cancellous regions of the distal femur and of muscle mass based on the soleus to body mass ratio. However, compared with the soleus, muscle mass or the muscle mass to body mass ratio of extensor digitorum longus (EDL) was not different between the control and HU groups. So, we think that pain-related behavior in our model participated more in bone than other tissues such as muscle.
On the comment of [In the materials and methods on line 235-238, the paragraph about the origin of the model should be deleted or moved to the introduction or discussion. Not relevant in this chapter.]
Reply: Thank you for suitable comment. We moved the paragraph to discussion of first paragraph.
Reviewer 2 Report
The authors should consider the followings: 1. The author should demonstrate how their treatment of anti-IL-6 receptor antibody could sufficiently block the IL-6. Any experimental evidences to show a decrease in IL-6 level? (protein level, gene level?) 2. How did the author determine the dose of anti-IL-6 receptor antibody used in the study? Please provide rationale(s) or experimental support for that. 3. Please give rationales of choosing the route of administration for anti-IL-6 receptor antibody. 4. Was the vehicle group in the animal model (vehicle of alendronate treatment, or vehicle of anti-IL-6 receptor antibody, or both)? Please explain if any missing vehicle group(s) being tested. 5. In Figure 2 and Figure 4, did the author practice blinding in counting (or assessing) the groups of different treatments? please explain why if that was not. 6. Please state clearly the novelty of this research in your abstract and conclusion. 7. Another set of higher magnification of groups in Figure 2A should be provided. 8. Please denote in Figure legend of Figure 1 (and other figures with statistical analyses) that the error bar represent SD (or SEM). 9. Improved figure resolution of Figure 2A should be considered. 10. A better figure legend by the graph axis should be observed, in Fig 1 and Fig 2. 11. The authors should consider using English proof-reading services by language professional.Author Response
Replies to Reviewer #2
The authors are grateful to Reviewer #2 for insightful and critical review comments that significantly help improve the manuscript and increase clinical relevance. Corrections in the newly revised manuscript are red color.
Please note that your review comments are shown in italic below and our replies in non-italic.
On the comment of [1. The author should demonstrate how their treatment of anti-IL-6 receptor antibody could sufficiently block the IL-6. Any experimental evidences to show a decrease in IL-6 level? (protein level, gene level?).]
Reply: Thank you for your valuable suggestion. In previous study, we investigated mRNA of IL-1, IL-6 and TNF of bone in hindlimb unloading mice. And these cytokine mRNA of hindlimb bone was elevated. Other study reported that the administration of anti-IL-6R antibody, MR16-1, inhibited the clearance of IL-6 due to IL-6R blockade. However, we have no data in this study whether treatment of anti-IL-6 receptor antibody could sufficiently block the IL-6 or not. We added the limitation that “We did not demonstrate how their treatment of anti-IL-6 receptor antibody could sufficiently block the IL-6” on Line 225-226.
On the comment of [2. How did the author determine the dose of anti-IL-6 receptor antibody used in the study? Please provide rationale(s) or experimental support for that.] and [3. Please give rationales of choosing the route of administration for anti-IL-6 receptor antibody.]
Reply: Thank you for suitable comment and suggestion. We determined the dose of anti-IL-6 receptor antibody, MR16-1, and chose the route of administration for it according to a previous report (reference 29).
On the comment of [4. Was the vehicle group in the animal model (vehicle of alendronate treatment, or vehicle of anti-IL-6 receptor antibody, or both)? Please explain if any missing vehicle group(s) being tested.]
Reply: Thank you for suitable comment and suggestion. We did not investigate the treatment of alendronate and anti-IL-6 receptor antibody in HL mice. We added the limitation on Line229-231, “And we did not investigate the treatment of vehicle group (HL mice). Future studies should address the dose- effects of treatment in hindlimb unloading-induced bone loss and effects of treatment in hindlimb bone of vehicle group.”
On the comment of [5. In Figure 2 and Figure 4, did the author practice blinding in counting (or assessing) the groups of different treatments? please explain why if that was not.]
Reply: Thank you for suitable comment and suggestion. We assessed the practice blinding in groups of different treatments. We added the following sentences “The immunostained sections were reviewed by each one observer who was blinded to the experimental group.” on Line340-341.
On the comment of [6. Please state clearly the novelty of this research in your abstract and conclusion.]
Reply: Thank you for suitable comment and suggestion. In this study, anti-IL-6 receptor antibody treatment prevented immobility-induced mechanical hyperalgesia in the hindlimbs of osteoporotic animal models and suppressed CGRP expression in DRG neurons. This treatment prevented the number of osteoclast cells, but not bone loss. The novelty of this research is that IL-6 is one of the causes of immobility-induced osteoporotic pain regardless improvement of bone loss. We added the novelty in abstract and conclusion.
On the comment of [7. Another set of higher magnification of groups in Figure 2A should be provided.] and [9. Improved figure resolution of Figure 2A should be considered.]
Reply: Thank you for suitable comment and suggestion. We provided the higher magnification of groups in CGRP expression in the DRG neurons in supplementary data (F).
On the comment of [8. Please denote in Figure legend of Figure 1 (and other figures with statistical analyses) that the error bar represent SD (or SEM).]
Reply: Thank you for suitable comment and suggestion. We changed the box plot graph in all figures for suggestion of Reviewer #1. We added the following sentence in each figure legend, “Top, bottom, and middle lines of the graph correspond to the 75th percentile, 25th percentile, and median, respectively. Cross represents mean. Each circle represents an outlier.”
On the comment of [10. A better figure legend by the graph axis should be observed, in Fig 1 and Fig 2.]
Reply: Thank you for suitable comment and suggestion. We added with the y-axis label in figure legend of Fig 1 and Fig 2.
Reviewer 3 Report
Manuscript (ijms-778665) describes interesting issues regarding the hindlimb suspension supplementation on piglets intestinal tract development, performance and health status.
However, the manuscript needs some corrections. Please, pay special attention to style correction – some sentences are constantly repeated, sometimes even in the same paragraph. This does not allow to read the work fluently, which reduces its value.
L 58 the end of the sentence/paragraph is missing.
The description of the procedures with von Frey filaments are insufficient.
Figure 1. How the force of 2 N can be obtained with 2g von Frey filaments? The value of 150% of response frequency does not make sense. Please remove. Maybe “frequency” is not the appropriate word.
L 109-111 and Supplementary data. How many mice were sacrificed at the start of the reload period (to preform uCT tomography)? Where did they come from? It's not written in the manuscript.
Figure 3. There is a difference between Hu vs ALN but not between HL vs HU or HL vs ALN? All the groups were the same size.
L275 correct to “BV/TV”
L281 Please describe TRAP staining in more detailed manner (reagents, producers, protocol). I was unable to find these information in [5, 26].
Supplementary data – please add the information about the number of animals in each examined group after reload (HL=8, HU=24, am I correct?).
Author Response
Replies to Reviewer #3
The authors are grateful to Reviewer #2 for insightful and critical review comments that significantly help improve the manuscript and increase clinical relevance. Corrections in the newly revised manuscript are red color.
Please note that your review comments are shown in italic below and our replies in non-italic.
On the comment of [L 58 the end of the sentence/paragraph is missing.]
Reply: Thank you for your valuable suggestion. This is a mistake of formatting changes to our original submission. We added the sentences “comparison with BP”.
On the comment of [The description of the procedures with von Frey filaments are insufficient.]
Reply: Thank you for your valuable suggestion. We added the procedures with von Frey filaments in details onLine 285-301.
On the comment of [Figure 1. How the force of 2 N can be obtained with 2g von Frey filaments? The value of 150% of response frequency does not make sense. Please remove. Maybe “frequency” is not the appropriate word.]
Reply: Thank you for suitable comment and suggestion of our mistake. We changed to “(g)” in Y axis of figure 1B. And we changed the box plot graph in all figures for suggestion of Reviewer #1. We changed the max value is 100% of response frequency in figure 1C and removed“frequency”.
On the comment of [L 109-111 and Supplementary data. How many mice were sacrificed at the start of the reload period (to preform uCT tomography)? Where did they come from? It's not written in the manuscript.]
Reply: Thank you for suitable comment and suggestion. No mice were sacrificed at the start of the reload period to preform uCT tomography. All mice were imaged under anesthesia to preform uCT tomography. We added the sentences “At the start of reload (after tail suspension for two weeks) mice were imaged under anesthesia, and at 2 weeks after reload (treatment) isolated femurs and tibias of sacrificed mice were imaged using a μCT scanner” on Line305-307.
On the comment of [Figure 3. There is a difference between Hu vs ALN but not between HL vs HU or HL vs ALN? All the groups were the same size.]
Reply: Thank you for suitable comment and suggestion. There is the difference between HL vs HU in BV/TV (bone volume/tissue volume), Tb.N (trabecular number) and Tb.Sp (trabecular separation) of distal femoral metaphysis, but not HL vs ALN in all analyses. We added the sentences “the differences between HL group and the ALN group were not significant in all analyses”, on Line139-140.
On the comment of [L275 correct to “BV/TV”.]
Reply: Thank you for suggesting our mistake. We corrected to BV/TV.
On the comment of [L281 Please describe TRAP staining in more detailed manner (reagents, producers, protocol). I was unable to find these information in [5, 26]]
Reply: Thank you for suitable comment and suggestion. We described TRAP staining in more detailed manner on Line317- 329.
On the comment of [Supplementary data – please add the information about the number of animals in each examined group after reload (HL=8, HU=24, am I correct?).]
Reply: Thank you for suitable comment and suggestion. The number of animals in each examined group after reload is correct, HL=8 and HU=24. At the start of reload (after tail suspension for two weeks) mice were imaged under anesthesia using a μCT scanner. We changed and added in detail on Line 303-313.
Round 2
Reviewer 1 Report
The manuscript has improved, but I still have some concerns, which remain unanswered.
Major concerns:
- The authors have fully addressed this concern, but the high magnification images should be included as an insert in Figure 2A.
- In figure 4, the provided images are unchanged and still fare from convincing. I don’t understand why osteoclast number is now changes to Oc.N/B.Pm, without any change in the values within the graphs. Have the authors just changes the unit, but not done the measurement of mm B.Pm investigated? Something just doesn’t add up. The M&M doesn’t help. Please provide the raw data for the mm B.Pm investigated and total number of osteoclasts investigated in each mouse.
- In general the style of graphs in non-classical and lacking basic information about the reported parameters that are only reported in the figure legends. This remains to be changes. The legend on the y-axis should include the parameter, not just the unit.
- Satisfactory addressed.
- Satisfactory addressed.
- Very fine.
- Box plot is acceptable, but not perfect.
- Indeed there is not direct measure of bone pain, only surrogate markers. Therefore one often include more than one surrogate marker to render it more likely that the surrogate marker reflect bone in the mice. As this is the case, the title should be changed to be less categorical on the bone pain. “Functional block of interleukin-6 reduce a bone pain marker, but not bone loss in hindlimb-unloaded mice”
- Very fine
Round 3
Reviewer 1 Report
The manuscript has improved, but I still have some concerns, which remain unanswered.
Point 1: I wrote that the high magnification images should be included as an insert in Figure 2A. It looks strange it include it as panel E. Then the authors should delete panel A, and show only the high magnification image of panel E as panel A. Not both, as done now.
Point 2: Very nice that the authors have corrected this mistake, but why have they provided two measurements for each animal in the table? One can only include one mean measurement for each animal in the statistical analysis. Please modify the stats accordingly.
Point 3-7: Very fine
Author Response
Replies to Reviewer #1
Authors are grateful to Reviewer #1 for insightful and critical review comments that significantly help improve the manuscript and increase clinical relevance. Corrections in the newly revised again manuscript are green color.
Please note that your review comments are shown in italic below and our replies in non-italic.
On the comment 1 of [I wrote that the high magnification images should be included as an insert in Figure 2A. It looks strange it include it as panel E. Then the authors should delete panel A, and show only the high magnification image of panel E as panel A. Not both, as done now.]
Reply: Thank you for your valuable pointing and suggestions. We deleted Figure 2A, and show only the high magnification image of Figure 2E as Figure 2A.
On the comment 2 of [Very nice that the authors have corrected this mistake, but why have they provided two measurements for each animal in the table? One can only include one mean measurement for each animal in the statistical analysis. Please modify the stats accordingly.]
Reply: Thank you for suitable comment. We modified the stats accordingly as the mean measurement for each animal in the statistical analysis.
On the comment 3-7
Reply: Thank you.